# Ectopic Expression of OsJAZs Alters Plant Defense and Development

**DOI:** 10.3390/ijms23094581

**Published:** 2022-04-21

**Authors:** Baolong Sun, Luyue Shang, Yang Li, Qiang Zhang, Zhaohui Chu, Shengyang He, Wei Yang, Xinhua Ding

**Affiliations:** 1State Key Laboratory of Crop Biology, Shandong Provincial Key Laboratory of Agricultural Microbiology, College of Plant Protection, Shandong Agricultural University, Tai’an 271018, China; sunbaolong1997@outlook.com (B.S.); shangluyue0523@163.com (L.S.); yangli1988@sdau.edu.cn (Y.L.); zqecnu@163.com (Q.Z.); 2State Key Laboratory of Hybrid Rice, College of Life Sciences, Wuhan University, Wuhan 430072, China; zchu77@whu.edu.cn; 3Department of Biology, Duke University, Durham, NC 27708, USA; shengyang.he@duke.edu; 4Key Laboratory of Quality Improvement of Agricultural Products of Zhejiang Province, College of Modern Agricultural, Zhejiang A&F University, Hangzhou 311300, China

**Keywords:** jasmonic acid, transcriptional repressors, JA insensitive, flowering, root growth, plant immunity

## Abstract

A key step in jasmonic acid (JA) signaling is the ligand-dependent assembly of a coreceptor complex comprising the F-box protein COI1 and JAZ transcriptional repressors. The assembly of this receptor complex results in proteasome-mediated degradation of JAZ repressors, which in turn bind and repress MYC transcription factors. Many studies on JAZs have been performed in *Arabidopsis thaliana*, but the function of JAZs in rice is largely unknown. To systematically reveal the function of OsJAZs, in this study, we compared the various phenotypes resulting from 13 *OsJAZs* via ectopic expression in *Arabidopsis thaliana* and the phenotypes of 12 *AtJAZs* overexpression (OE) lines. Phylogenetic analysis showed that the 25 proteins could be divided into three major groups. Yeast two-hybrid (Y2H) assays revealed that most OsJAZ proteins could form homodimers or heterodimers. The statistical results showed that the phenotypes of the *OsJAZ* OE plants were quite different from those of *AtJAZ* OE plants in terms of plant growth, development, and immunity. As an example, compared with other *JAZ* OE plants, *OsJAZ11* OE plants exhibited a JA-insensitive phenotype and enhanced resistance to *Pst* DC3000. The protein stability after JA treatment of OsJAZ11 emphasized the specific function of the protein. This study aimed to explore the commonalities and characteristics of different JAZ proteins functions from a genetic perspective, and to screen genes with disease resistance value. Overall, the results of this study provide insights for further functional analysis of rice JAZ family proteins.

## 1. Introduction

The oxylipin-derived phytohormone jasmonic acid (JA) is involved in regulating plant growth, development, secondary metabolism, and stress responses during their life cycle [1]. JAZ family proteins act as repressors of JA signaling. Current models of JA signaling indicate that JA regulates various biological processes via the COI1-JAZ-MYC signaling pathway [2,3]. In the absence of JA, JAZ proteins bind to MYC2 transcription factors through their Jas domain and repress the transcription of downstream JA-related genes; JAZ proteins are degraded by COI1-mediated ubiquitination in the presence of the active JA form. This degradation releases MYC2, which in turn activates transcription [4,5].

The JAZ protein family belongs to the plant-specific TIFY superfamily, the members of which have two highly conserved functional domains: TIFY (also known as ZIM) and Jas (also known as cct_2) domains. Both the TIFY and Jas domains are required for the integral function of JAZ proteins in the JA signaling pathway. The TIFY domain usually consists of 28 amino acids and contains a highly conserved TIFY motif (TIF[F/Y]XG) [6]. The TIFY domain mediates hetero- and homomeric interactions between JAZ proteins [7,8]. In addition, the interaction between JAZ proteins and the NINJA-TPL complex is dependent on the TIFY domain [9]. The Jas domain, which is close to the C-terminus and consists of 29 amino acids with a conserved motif (SLX2FX2KRX2RX5PY), is specific to JAZ members [10,11,12]. The Jas domain functions as a connecter to mediate direct interactions with both COI1 and basic helix-loop-helix (bHLH) transcription factors, which results in the inhibition of transcriptional activity. A lack of the Jas domain was shown to cause both insensitivity to bioactive JA and constitutive repression of JA signaling [13,14,15].

The ZIM domain is present in the proteins of both higher and lower (e.g., moss) plants, suggesting an important role for JAZ in the evolution of terrestrial plants [6,16]. The role of the JAZ protein in growth has been widely reported in a variety of plant species. For example, *TaJAZ1* modulates seed germination and negatively regulates the expression of abscisic acid (ABA)-responsive genes in bread wheat [17]; NaJAZh-deficient plants show reduced nicotine levels and spontaneous necrotic damage to their leaves [18]; and overexpression of *SlJAZ2* alters tomato plant architecture at an early stage and accelerates flowering [19]. JAZ proteins also play an important role in plant defense. Overexpression of *GsTIFY10* in *Arabidopsis* enhances plant tolerance to bicarbonate stress at the seedling stage [20]; overexpression of *TaJAZ1* in bread wheat increases the accumulation of reactive oxygen species, enhancing resistance to powdery mildew [21]; and, in maize, ZmJAZ14 (ZmTIFY19) acts as a transcriptional regulator, while the expression of *ZmJAZ14* in *Arabidopsis thaliana* increases the tolerance of *Arabidopsis* to polyethylene glycol (PEG) stress, JA, and ABA [22].

JAZ proteins play two mutually restrictive and highly conserved roles: they act as transcriptional repressor and perceive signals, thus coordinating the balance between development and defense. In *Arabidopsis thaliana*, based on the redundancy of *JAZ* genes [23], researchers generated single, -double, and higher-order mutants to explore the function of *JAZ* genes [24,25]. However, the specificity and redundancy of *JAZ* genes are still not fully understood. In addition, some of the *AtJAZ*-overexpressing lines exhibited no distinctly different phenotypes. Therefore, we transformed *OsJAZ* genes into *Arabidopsis thaliana* to avoid functional redundancy and expected to obtain a significantly distinct phenotype.

Rice is one of the most important crop species worldwide and functions as a model organism of monocots. Some studies have shown the important role of JA in the defense response of rice. Unlike in *Arabidopsis*, only a few of the 15 *JAZ* genes in the rice family have been studied. For example, OsJAZ1 regulates spikelet development and acts as an OsBHLH148-related JA signal regulator to induce drought resistance in plants [26,27]; by functioning as a heterodimer with other JAZ proteins to repress JA signaling in rice, OsJAZ8 plays an important role in resistance to Xoo (*Xanthomonas oryzae* pv. *oryzae*) [28]; OsJAZ9 interacts with OsNINJA and OsbHLH062 to regulate the expression of JA response genes involved in rice salt tolerance [29,30]; and OsJAZ13 regulates the expression of genes related to the JA/ethylene response and activates hypersensitive cell death in rice by recruiting the cosuppressor OsTPL [31]. In conclusion, JAZ proteins play an important role in regulating plant growth, defense, and stress responses. However, research on JA signal-mediated defense involving JAZ proteins in rice is still lacking. By analyzing 13 *OsJAZ-* and 12 *AtJAZ*-overexpressing *Arabidopsis* plants, we found that different *JAZ* genes lead to different phenotypes in terms of pathogen defense, root growth, and flowering time. Moreover, the OsJAZ11 family was found to be the most stable JAZ family in rice. These results lay the foundation for further elucidation of the functions of *JAZ* genes and their regulatory mechanisms.

## 2. Results

### 2.1. Phylogenetic Analysis of OsJAZs and AtJAZs

To analyze the evolutionary relationships of JAZ proteins, a phylogenetic tree was generated on the basis of the sequence alignments of 25 full-length JAZ proteins, 13 from rice, and 12 from *Arabidopsis thaliana*. The 25 JAZ proteins could be classified into three major groups according to the unrooted phylogenetic tree (Appendix A). The results showed that the rice JAZ family members were homologous to the *Arabidopsis* JAZ family members. For example, OsJAZ5 and AtJAZ10 clustered together on the smallest branch of the evolutionary tree, suggesting that they may play similar roles in different plant species.

Nine *OsJAZs* (*OsJAZ5–13*) have been reported to be induced by wounding or JA treatment [29]. They were all clustered in group I except OsJAZ5, which was in group II. All the *AtJAZ* genes except *AtJAZ11* have been shown to be induced after insect feeding, and *AtJAZ1*, *AtJAZ2*, *AtJAZ5*, *AtJAZ6*, *AtJAZ9*, *AtJAZ10*, and *AtJAZ12* showed relatively high levels of accumulation [32] and were also clustered into group I and group II. *OsJAZ10* and *OsJAZ13* have similar induction dynamics [29], and they clustered together on the smallest branch of the evolutionary tree. *OsJAZ2* is the only gene that can be induced by ABA and was clustered in group III, independent of other *JAZ* genes [29]. *JAZ* genes with the same motif clustered more closely in the evolutionary tree and had more similar expression patterns.

### 2.2. The Majority of OsJAZs Affect Flowering Time via Coordination with AtFT Expression

JA regulates a wide range of developmental processes, including flowering time. To study the effect of ectopic expression of *OsJAZs* on flowering, we overexpressed 13 OsJAZ-GFP fusion proteins in *Arabidopsis thaliana* under the control of the 35S promoter. Three to five different lines were made for each overexpressing (OE) plants. Western blot analysis showed that each fusion protein had similar expression levels in 13 *OsJAZ* OE plants (Appendix A). The 12 *AtJAZ* OE plants were generated in our laboratory during a previous study [33]. We compared the flowering phenotypes of 13 *OsJAZ* OE plants and 12 *AtJAZ* OE plants under long-day (LD) conditions (Figure 1A and Appendix A). Among the 13 *OsJAZ* OE plants, the *OsJAZ3*, *-4*, *-5*, *-8*, *-9*, *-10*, and *-11* OE plants exhibited a significantly early-flowering phenotype. *OsJAZ1*, *OsJAZ4-2* (an alternatively spliced (AS) transcript of OsJAZ4), and *OsJAZ12* OE exhibited delayed flowering. Among the 12 *AtJAZ* OE plants, the *AtJAZ1*, *-2*, *-3*, *-4*, *-7*, *-8*, *-9*, *-10*, and *-11* OE plants exhibited a significantly early-flowering phenotype. We also counted the number rosette leaves when the plants bloomed. Compared with the later-flowering lines, the earlier-flowering lines had fewer rosette leaves (Figure 1B). When ectopically expressed in *Arabidopsis thaliana*, most *OsJAZ* genes promote plant flowering, similar to *AtJAZ* genes. However, several *OsJAZ* genes delay plant flowering, implying that these genes play different functions in flowering.

Plant flowering is a complex process influenced by photoperiod, temperature, and stress. FT (Flowering locus T) is one of the most important floral integrators regulated by several signaling pathways. The transfer of FT from leaves to the shoot apical meristem initiates the flowering process in plants [34,35,36]. Many studies have demonstrated that early flowering is often accompanied by upregulated expression of *FT* or its homologous genes [37,38,39]. Thus, we measured the expression levels of *FT* in all our transgenic plants (Figure 1C) and correlation analysis was carried out to evaluate the relationship between *FT* expression and flowering time. As shown in Appendix A, the expression level of *FT* and the flowering time were significantly negatively correlated (r = −0.4466, *p* < 0.05). This indicated that OsJAZs and AtJAZs perform similar functions in *Arabidopsis* flowering and that the participation of OsJAZs in flowering time is also related to the expression of *AtFT*. A previous study showed that AtJAZs interact with DELLA proteins to affect the gibberellic acid (GA) signaling pathway and thus affect flowering time [33]. We speculated that this regulatory mechanism is conserved when *OsJAZs* are ectopically expressed in *Arabidopsis thaliana*. We chose OsJAZ9, -10, -11, -12, and -13 for Y2H analysis because they clustered closely together according to evolutionary analysis, but their OE lines exhibited different flowering phenotypes. The results showed that OsJAZ9 and OsJAZ11 directly interact with the DELLA protein GAI, which is consistent with the early-flowering phenotypes of those plants (Figure 1D). There was no interaction between OsJAZ10, -12, or -13 and the two DELLA proteins, and overexpression of *OsJAZ12* and *OsJAZ13* resulted in delayed flowering or no significant differences compared with the phenotype of ecotype Columbia (Col-0) plants. Although we did not detect any interaction between OsJAZ10 and these two DELLA proteins (Figure 1D), *OsJAZ10* OE plants still flowered early, probably due to altered signal transduction.

### 2.3. OsJAZs Regulate Root Growth in Arabidopsis

Abscisic acid (ABA) can inhibit the germination and post-germination growth of *Arabidopsis thaliana* seeds, and jasmonic acid (JA) can enhance the function of ABA [40]. Exogenous application of methyl jasmonate (MeJA) has been shown to inhibit various parts of seedling growth, including root growth and hypocotyl elongation [9,41,42]. To determine the response of *OsJAZ* genes to exogenous JA, we compared the root growth inhibition among the 13 *OsJAZ* OE and 12 *AtJAZ* OE plants with or without exogenous MeJA application. When the transgenic plants were planted on 1/2-strength Murashige and Skoog (MS) media without MeJA, statistical analysis showed that the *OsJAZ3*, *-6*, *-8*, *-9*, *-10*, and *-11* OE plants had significantly shorter roots than the wild-type plant, but the *OsJAZ4*, *-4-2*, and *-5* OE and the *AtJAZ9,* and *-12* OE plants had significantly longer roots. When the transgenic plants were under 50 µM MeJA treatment, the *OsJAZ1*, *-6*, *-8*, *-11*, *-12*, -*13* OE plants and the *AtJAZ5*, *-6*, *-8*, *-11*, *-12* OE plants had longer roots than the wild-type plants, but the roots of the *OsJAZ3*, *-4*, *-4-2*, *-5*, and *-7* OE plants were shorter (Figure 2A and Appendix A).

Based on the root length of transgenic plants and Col-0 on 1/2-strength MS media with or without MeJA, we calculated their resistance to MeJA (Figure 2B). All *OsJAZ* OE plants were significantly different to Col-0, the MeJA resistance of *OsJAZ1*, *-6, -8, -9, -10 -11, -12, -13* OE plants were significantly higher than the wild-type, while the *OsJAZ3*, *-4*, *-4-2*, *-5*, and *-7* OE plants were lower. Most *AtJAZ* OE plants have similar resistance to Col-0 except *AtJAZ5*, *-6*, *-8* have higher resistance than Col-0. Although *AtJAZ9* OE is also significantly different from Col-0, the difference is too small (about 2%) to be included in the discussion. Notably, the MeJA resistance of *OsJAZ11* OE plants was the highest among those of all the Os*JAZ* OE plants, implying that OsJAZ11 is the most effective JA repressor of the OsJAZ family.

### 2.4. Various OsJAZs Are Involved in Plant Immunity, and Overexpression of Only OsJAZ11 Enhances Plant Resistance to Pst DC3000

To determine the effect of OsJAZs on pathogen defense, we inoculated the OE plants with *Pst* DC3000. Compared with the wild type, many transgenic plants showed decreased bacterial resistance (Figure 3A). In general, the *OsJAZ* OE plants showed lower resistance to *Pst* DC3000 than *AtJAZ* OE plants did. This finding underscores the potential importance of JA signaling in pathogen defense in plants. Interestingly, compared to the wild-type and other *OsJAZ* OE plants, the *OsJAZ11* OE plants showed strong resistance to *Pst* DC3000.

Previous research has shown that the SA pathway is involved in resistance against *Pst* DC3000, and JAZ family proteins act as repressors of JA pathway. Based on the antagonism of JA/SA, we speculate that the resistance of *OsJAZ11* OE to *Pst* DC3000 may be due to the changes of SA levels in plants. Thus, the expression level of SA response gene *PR1* was analyzed in the OE lines. We chose *OsJAZ9-13* OE plants for this analysis because their resistance to *Pst* DC3000 is representative. The *PR1* level of *OsJAZ11 OE* was significantly upregulated post-inoculation, and it was the highest among the Col-0 and *OsJAZ9-13* OE plants (Figure 3B). We also measured the expression level of the JA response gene *PDF1.2*. Notably, the *PDF1.2* level in *OsJAZ11* OE was much lower than that in the other lines and did not significantly change after inoculation (Figure 3C). These results suggest that OsJAZ11 is a strong JA repressor.

To test the possibility that the resistance of *OsJAZ11* OE plants to *Pst* DC3000 was not due to the insertion site, we inoculated five different lines of *OsJAZ11* OE plants with *Pst* DC3000. As shown in Figure 4B, all the transgenic lines showed enhanced resistance to *Pst* DC3000, and we found that all five *OsJAZ11* OE lines were slightly smaller than Col-0 (Figure 4A). This result confirmed that the resistance phenotype was due to *OsJAZ11* overexpression.

### 2.5. The Majority of OsJAZs Can Form Homodimers and Heterodimers

To determine the molecular mechanism of the OsJAZ phenotype, a yeast two-hybrid (Y2H) assay was used to determine the homomeric interactions of the 13 members of the OsJAZ family. We fused each full-length OsJAZ protein with a LexA BD and AD. Homomeric interactions were subsequently determined by examining the growth of yeast on SD/-Leu/-Trp/-His/-Ade media. As shown in Table 1, OsJAZ4-2, OsJAZ7, OsJAZ9, OsJAZ10, and OsJAZ11 strongly interacted as homodimers; however, other OsJAZ family members (OsJAZ1, OsJAZ3, OsJAZ4, OsJAZ5, OsJAZ6, OsJAZ8 and OsJAZ12) were unable to form homodimers.

By using Y2H analysis, we systematically tested all possible OsJAZ-OsJAZ interactions in both BD/AD orientations. Fifty-six interactions with different intensities were observed among the 169 possible combinations involving all family members (Table 1; Appendix A). We detected five interactions (OsJAZ1-OsJAZ11, OsJAZ4-OsJAZ10, OsJAZ4-2-OsJAZ7, OsJAZ4-2-OsJAZ11, OsJAZ4-2-OsJAZ12, OsJAZ6-OsJAZ11, OsJAZ7-OsJAZ11 and OsJAZ10-OsJAZ13) in each of the BD and AD orientations.

Next, we verified the authenticity of the OsJAZ-OsJAZ interactions in vivo using a bimolecular fluorescence complementation (BiFC) assay. We chose OsJAZ6, OsJAZ7 with OsJAZ11 for this analysis and OsJAZ9 with OsJAZ11 as a negative control. N-terminal fragments of yellow fluorescent protein (YFP) were fused to full-length OsJAZ6, OsJAZ7, and OsJAZ9 sequences, and the C-terminal YFP fragment was fused to OsJAZ11. Three possible OsJAZ fusion combinations of NYFP and CYFP were subsequently coexpressed in tobacco epidermal cells by using the *Agrobacterium*-mediated method. The results are shown in Figure 5. Bright fluorescence can be observed in the nucleus, suggesting that the OsJAZ proteins can indeed form heterodimers in vivo, the interactions of which occur in the nucleus.

### 2.6. OsJAZ11 Functions in the Nucleus

Previous studies have shown that JAZs are located in the nucleus [10,11]. We used confocal laser microscopy to determine the subcellular localization of the OsJAZ11-GFP fusion protein in the epidermal cells of transgenic *Arabidopsis* plants, and chose OsJAZ12-GFP, OsJAZ13-GFP as controls. The results showed that OsJAZ11-GFP, OsJAZ12-GFP, and OsJAZ13-GFP were predominantly located in the nucleus (Figure 6).

### 2.7. Different JAZ Proteins Show Different Sensitivity to JA

To further verify that there are differences in the response intensity of different OsJAZ proteins to JA, we monitored the GFP fluorescence changes in the roots of MeJA-treated transgenic seedlings to investigate the hormone-dependent stability of OsJAZs. We use *OsJAZ9-13* OE for this analysis because they have significant differences in root length (Figure 2A) and bacterial resistance (Figure 3A). As shown in Figure 7, a nuclear GFP signal was clearly observed in the absence of MeJA. In the 1 µM MeJA treatment group, fluorescence was not detected in *OsJAZ10* OE, *OsJAZ12* OE, and *OsJAZ13* OE roots and was diminished in *OsJAZ9* OE roots, but bright fluorescence was still observed in *OsJAZ11* OE roots. In the 100 µM MeJA treatment group, fluorescence not detected in any of the roots.

Taken together, these results indicated that different JAZ proteins show different sensitivity to JA, and OsJAZ11-GFP was the most stable protein in response to MeJA treatment among OsJAZ9-13. This also explains why, after exogenous MeJA treatment, the roots of *OsJAZ11* OE plants were longer than those of other *OsJAZ* OE plants.

## 3. Discussion

Members of the JAZ family are widely distributed in both lower and higher plants [43,44,45]. Previous studies have examined the growth, development, and stress resistance of several OsJAZs [27,28,29,30,31] and AtJAZs [11,46,47,48,49]. In this study, we compared 13 *OsJAZ* OE plants and 12 *AtJAZ* OE plants for the first time and analyzed the function of each JAZ protein in the same plant genetic background. According to the evolutionary tree we constructed, the genes that were closely clustered together were also those whose overexpression resulted in similar phenotypes (Table 2). We found that the flowering period, root length, and immunity of some *JAZ* OE plants are different, and there were differences between *OsJAZ* OE and *AtJAZ* OE plants. We also preliminarily analyzed the underlying mechanisms responsible.

A few studies have mentioned that OsJAZs are involved in plant flowering [40]. Interestingly, in contrast to *OsJAZ3*, *-4*, *-5*, *-8*, *-9*, *-10*, and *-11* OE plants, which showed early flowering, *OsJAZ1*, *-4-2*, and *-12* OE plants showed delayed flowering, which differed from that exhibited by the *AtJAZ* OE plants (Figure 1A). This suggests that there may be a JAZ regulatory mechanism in rice that is different from that in *Arabidopsis*. When plants perceive stress, endogenous JA promotes the degradation of JAZs, thereby relieving the transcriptional inhibition of TARGET OF EAT1 (TOE1) and TOE2. The released TOEs inhibit the transcription of *FT* and trigger a signaling cascade to delay flowering [50]. AtJAZ1, -3, -4, and -9 have been proven to interact with AtTOE1/2, and *35S pro:JAZ1∆Jas* flowers earlier [51]. There are also articles showing that JAZs interact with DELLA proteins to hinder DELLA-PIF interactions to regulate growth phenotypes, including flowering time [52]. AtJAZ1, -3, -4, -9, -10, and -11 have been shown to interact with DELLA proteins, and *AtJAZ1*, *-3*, *-4*, *-9*, *-10*, and -*11* OE plants show early-flowering phenotypes, while *AtJAZ5* and *-6* OE plants do not [33]. In this article, we found that the *AtJAZ1*, *-2*, *-3*, *-4*, *-7*, *-8*, *-9*, *-10*, and -*11* OE plants showed early flowering, while the *AtJAZ5*, *-6*, and *-12* OE plants did not (Figure 1A), and the transcript level of *FT* was consistent with the phenotype (Figure 1C). Plant flowering is a complex process that is regulated by multiple factors and signaling pathways [53,54]. Therefore, we speculate that, in addition to the above two signaling pathways, JAZ proteins can also affect flowering through other pathways.

In addition to flowering time, compared with the *AtJAZ* OE plants, the *OsJAZ* OE plants showed a more significant phenotype in terms of their JA response and root length, indicating that ectopic expression of *OsJAZ* could partly avoid functional redundancy among *OsJAZs.* It has been reported that the inability to degrade OsJAZ1 can prevent the inhibition of plant root length by JA [26]. This finding is consistent with the phenotype of the *OsJAZ1* OE plants in this study. The degradation of OsJAZ8 induced by JA depends on its jas domain, so the root length of *OsJAZ8ΔC* OE rice plants is hardly inhibited by MeJA [28]. In this study, compared with wild type plants, the *OsJAZ8* OE plants were more resistant to MeJA. Rice plants overexpressing *OsJAZ13* showed several JA-insensitive phenotypes [31], which was consistent with *OsJAZ13 OE* in this study (Figure 2A). After JA induction or wounding, the transcript levels of *OsJAZ9*, *OsJAZ11* and *OsJAZ13* were particularly significant (more than 100-fold) [29]. In the present study, the root tips of the *OsJAZ9* OE and *OsJAZ11* OE plants showed more significant fluorescence than did those of other *OsJAZ* OE plants after 1 µM MeJA treatment (Figure 7). In addition, *OsJAZ9* OE, *OsJAZ11 OE* and *OsJAZ13 OE* showed higher MeJA resistance during root development (Figure 2B). Taken together, these results demonstrate the credibility of our research.

After we calculate the resistance of *JAZ* OE to MeJA, we found that overexpression of OsJAZ proteins in phylogenetic tree group I resulted in increased MeJA resistance in *Arabidopsis* plants, except for OsJAZ7; overexpression of OsJAZ proteins in phylogenetic tree group II resulted in decreased MeJA resistance, except for OsJAZ1. AtJAZ proteins, which resulted in increased MeJA resistance in *Arabidopsis thaliana* plants after overexpression, were also not located in group II (Table 2). These results suggest that JAZ proteins in phylogenetic tree group II may have or lack a specific domain that allows them to have this property. Our future research will focus on this question.

The resistance of most *JAZ* OE plants decreased significantly compared with that of Col-0 post-*Pst* DC3000 inoculation. In general, compared with the *AtJAZ* OE plants, the *OsJAZ* OE plants showed significantly lower bacterial resistance. Unlike *OsJAZs*, among the 12 *AtJAZ* OE lines, there were no lines with increased disease resistance. The resistance of *OsJAZ11* OE to *Pst* DC3000 significantly improved (Figure 3). Interestingly, plants that overexpressed genes from the same cluster showed the opposite phenotype. This phenomenon was observed for flowering time, root length, sensitivity to JA, and immunity. Moreover, Y2H analysis demonstrated widespread and sophisticated interactions among the OsJAZ family members (Table 1). Taken together, these findings indicate that there may be different JAZ regulatory networks in rice and *Arabidopsis*.

When the transgenic plants were transplanted onto 1/2-strength MS media supplemented with 50 µM MeJA, the root length of the *OsJAZ11* OE plants was the longest of all *OsJAZ* OE plants (Figure 2A). After the root tips of the transgenic seedlings were treated with 1 µM MeJA for two hours, the fluorescence of *OsJAZ11* OE was the strongest among the transgenic plants we tested (Figure 7). These results suggest that the OsJAZ11-GFP fusion protein is the most stable fusion protein under MeJA processing. Among all the *JAZ* OE lines, only *OsJAZ11* OE showed increased resistance to *Pst* DC3000 (Figure 3). Previous research has shown that the SA pathway is involved in resistance against *Pst* DC3000 [55,56,57,58]. Furthermore, *Pst* DC3000 produces coronatine, a structural mimic of bioactive JA that is more potent than jasmonates in activating JA responses [4] and coronatine also downregulates SA response [59,60,61,62]. By overexpressing the strong and stable repressor OsJAZ11, these plants are presumably able to withstand degradation of the JAZs after coronatine exposure [63]. Therefore, JA response is not increased (as shown by low *PDF1.2* levels in Figure 3C), and the plant is able to induce the SA response (marked by increased *PR1* shown in Figure 3B) and mounts a defense response that results in increased resistance to *Pst* DC3000.

Instead of an analysis of *OsJAZ* transcript levels via “omics” methods in different growth states or under stress, we performed an intuitive visual analysis from a genetic perspective. This work is an exploration of the common features and characteristics of *OsJAZ* genes, as well as a supplement to the studied genes. JA plays an important role in resisting insect feeding and necrotic pathogens and coordinating the balance between growth and development together with other hormones. The strong JA repressor OsJAZ11 we identified can be used as a reference for future plant breeding.

## 4. Materials and Methods

### 4.1. Construction of Vectors

The vectors for gene transformation and components used for Y2H assays were constructed via the Gateway system. The following coding sequences were first cloned into the entry vector pDONR221: *OsJAZ1*, *-3*, *-4*, *-4-2*, *-5*, *-6*, *-7*, *-8*, *-9*, *-10*, *-11*, *-12*, and -*13*. Afterward, they were inserted into the vector pEarleyGate-103 to construct transgenic plants or inserted into both the pDEST-GADT7 and pDEST-GBKT7 vectors for Y2H analysis. Other Y2H analysis and transient expression vectors were constructed with restriction enzyme digestion and recombinant ligation. The coding sequences of *AtMYC2*, *AtGAI*, and *AtRGA* were cloned and inserted into the pGADT7-AD vector for Y2H analysis. The coding sequence of *OsJAZ11* was cloned and inserted into the pSPYCE vector, and the coding sequences of *OsJAZ6*, *-7*, and *-9* were cloned into pSPYNE vectors for BiFC analysis. The primers used are listed in Appendix A.

### 4.2. Plant Material and Growth Conditions

*Arabidopsis thaliana* Col-0 was used as the wild type for all the experiments. Twelve *AtJAZ* OE plants were from our previous work [33]. After dark treatment at 4 °C for 2–3 days, seeds of *Arabidopsis thaliana* were evenly and randomly sown onto 1/2-strength MS media. They were subsequently transferred to a growth chamber at 22 °C under 16 h light (100 μE·m^−2^·s^−1^) and 8 h of darkness. For root length growth inhibition assays, seeds were grown in 1/2-strength MS media for 10 days. For other assays, after the seeds germinated and the seedlings had two cotyledons, they were transferred to vermiculite.

### 4.3. Construction of Transgenic Plants

Transgenic *Arabidopsis* lines were prepared by the floral dip transformation method [64] and selected on 1/2-strength MS plates containing basta (50 mg·mL^−1^). The phenotypes of the transgenic plants were verified in at least three independent transgenic lines. Homozygous lines were identified in the T3 generation on the basis of their resistance to basta.

### 4.4. Phylogenetic Analysis

MEGA 7 (https://www.megasoftware.net/) was used to perform multiple alignments of the amino acid sequences of the Jaz family members. A phylogenetic tree was constructed using the neighbor-joining method in MEGA 7 software [65,66]. The bootstrap value was set to 1000 repeats, which was used to detect the confidence of each part of the phylogenetic tree [67,68,69].

### 4.5. Quantification of Flowering Time

Seedlings and plants were grown until they flowered inside a growth chamber under LD conditions (16 h of light:8 h of darkness). More than 20 plants of each genotype were planted for each independent experiment. Flowering time was scored as the number of total rosette leaves at bolting and the number of days from germination to flowering (three biological replicates each).

### 4.6. Y2H Assays

Y2H assays were performed by using the BD Matchmaker system (Clontech, Palo Alto, CA, USA). The PCR products of the *OsJAZs* and *AtJAZs* coding DNA sequences (CDSs) were digested, subcloned, and inserted into the pGADT7 and pGBKT7 vectors, respectively. The yeast strain Y2HGold, which harbors the chromosomally integrated reporter genes lacZ, which encodes β-galactosidase, and HIS, which mediates histidine biosynthesis under the control of the GAL1 promoter, was used for transformation of the expression constructs. Transformation of Y2HGold cells was performed according to the manufacturer’s instructions. SD/TL—(lacking tryptophan and leucine) and SD/TLHA—(lacking tryptophan, leucine, histidine, and adenine) were used.

### 4.7. JA Resistance Analysis

Percentage of resistance to MeJA for each overexpressing plant and Col-0 were generated by calculating the ratio of the root length of per *Arabidopsis* grown on 1/2 MS media supplemented with 50 µM MeJA to per *Arabidopsis* grown on 1/2 MS media without MeJA.

### 4.8. Pathogen Infection Assays

Four-week-old plants grown in the soil under LDs at 22 °C were used for pathogen infection and gene expression assays. Bacterial cells of the virulent *Pst* DC3000 strains were cultured for 24 h at 28 °C in potato-saccharose agar media supplemented with rifampicin (50 mg·L^−1^) to an OD_600_ of 0.8–1.0. A bacterial cell suspension was prepared at OD_600_ = 0.002 in 10 mM MgCl_2_ and pressure infiltrated into the 4th and 5th rosette leaves. The inoculated plants were grown in a growth chamber at 22 °C with a relative humidity of 80% for 2 days under LD conditions before bacterial cell counting.

### 4.9. Gene Transcription Assays

Total RNA was extracted using TRIzol reagent (Invitrogen, Carlsbad, USA) following the manufacturer’s instructions. Reverse transcription reactions were performed using HiScript II Q RT SuperMix for qPCR (+gDNA wiper) (VAZyme, Nanjing, China), and quantitative PCR was performed with Ultra SYBR Mixture (Low ROX) (CWBIO, Beijing, China) on an IQ5 Real-Time PCR System (Bio-Rad, Hercules, CA, USA). The expression values for each gene were normalized to ACTIN2 (AT3G18780) expression according to the 2^−ΔΔCt^ method. The detailed method is described previously, and the primers used are listed in Appendix A.

### 4.10. BiFC Assays

Full-length *OsJAZ6*, *-7*, *-9* and *-11* CDSs were cloned and inserted into pSPYNE and pSPYCE, respectively. The primer information is shown in Appendix A. The obtained vector was transformed into *Agrobacterium tumefaciens* GV3101. The two *Agrobacterium* mixtures were separately mixed to verify the interaction in equal amounts. Then, the mixture was injected into the abaxial side of the tobacco leaves so that the whole leaf was filled with water stains [70]. After the injection, the tobacco was placed under continuous light at 26 °C for at least 50 h. The epidermis of the injection site was cut and removed to make a temporary slide and placed under an LSM 880 confocal microscope (Carl Zeiss, Oberkochen, Germany) for observations.

### 4.11. Subcellular Protein Localization

Live plant imaging was performed on LSM 880 confocal microscope using. Fluorescence intensity analyses were performed using Zen evaluation software (Carl Zeiss, https://www.zeiss.com.cn/). To visualize the nuclei, leaves of 5-week-old transgenic *Arabidopsis* plants were infiltrated via a syringe 2 h prior to imaging with a solution that included 10 mg·mL^−1^ 4′,6-diamidino-2-phenylindole (DAPI; Sigma-Aldrich, St. Louis, MO, USA). GFP and DAPI fluorescence were monitored simultaneously by excitation at 514 nm (argon laser) and 405 nm (diode laser), respectively. The GFP and DAPI fluorescence was detected after passage through 530- to 600-nm bandpass and 474- to 525-nm emission filters.

### 4.12. In Vivo OsJAZ-GFP Degradation Assays

The OsJAZ9-13 was transgenic plants were cultivated on 1/2-strength MS media for seven days under LD conditions. After treatment with 0, 1, and 100 mM MeJA for 2 h, the fluorescence of the OsJAZ-GFP fusion protein was observed under an LSM 880 confocal microscope [7]. All the images were taken under the same exposure time and software settings.

## 5. Conclusions

Our study aimed to explore the commonalities and characteristics of different JAZ proteins functions from a genetic perspective, and to screen genes with disease resistance value. Our above results proved that the overexpression of major JAZ proteins altered the flowering period (Figure 1), tolerance to JA (Figure 2) and bacterial resistance (Figure 3) of *Arabidopsis* to varying degrees. Among them, *OsJAZ11* OE showed an early flowering phenotype, had strong JA resistance, and interacted with various OsJAZ proteins (Table 1). The bacterial resistance of *OsJAZ11* OE is significantly improved through the antagonism of SA and JA. These findings demonstrate that OsJAZ11 may play an important role in the JA signaling pathway and plant immune responses.

## Figures and Tables

**Figure 1 ijms-23-04581-f001:**
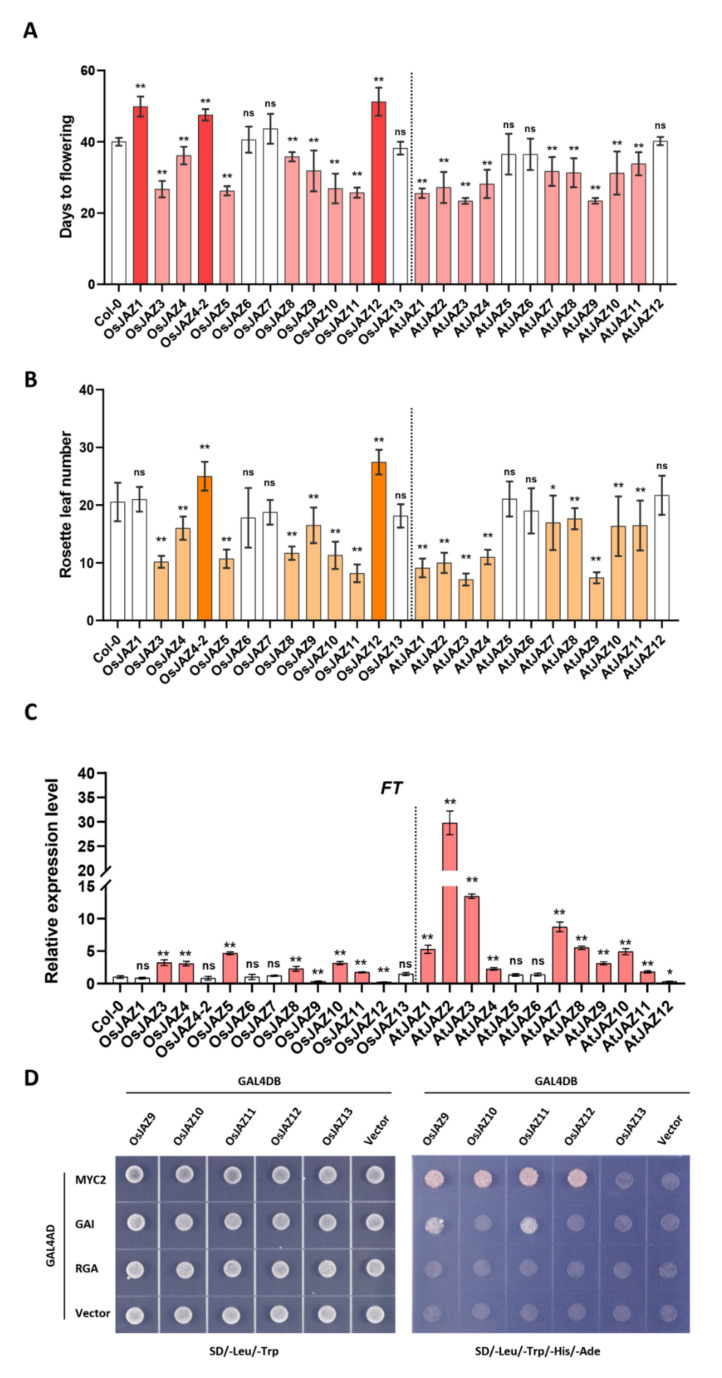
Function of JAZs in flowering phenotypes. (**A**) plant flowering time. The dark red column represents that the transgenic line flowered later than Col-0, the light red column represents that the transgenic line flowered earlier than Col-0, and the white column represents that the flowering time of the transgenic line is not significantly different from that of Col-0. The data shown are the means ± SEs (*n* = 12) (*, *p* < 0.05; **, *p* < 0.01; ns, no significance; Student’s *t*-test); (**B**) number of rosette leaves when the plants were blooming. The dark orange column represents that the number of rosette leaves of the transgenic line is more than that of Col-0, the light orange column represents that the number of rosette leaves is less than that of Col-0, and the white column represents that there is no significant difference in the number of rosette leaves of the transgenic line compared with Col-0. The results are shown as the means ± SEs (*n* = 8 to 12) (*, *p* < 0.05; **, *p* < 0.01; ns, no significance; Student’s *t*-test). (**C**) *FT* expression in the indicated transgenic plants. The leaves of 10-day-old plants were collected, and total RNA was extracted for qRT-PCR analysis. The light red column represents a significant difference in the *FT* expression of the transgenic line compared with Col-0, and the white column represents no significant difference compared with Col-0. The above values are the means ± SEs (*n* = 3) (*, *p* < 0.05; **, *p* < 0.01; ns, no significance; Student’s t test). Three biological repeats with similar results were included. (**D**) Y2H assay showing that OsJAZ9, -10, -11, and -12 interact with AtMYC2 and that OsJAZ9 and OsJAZ11 interact with AtGAI.

**Figure 2 ijms-23-04581-f002:**
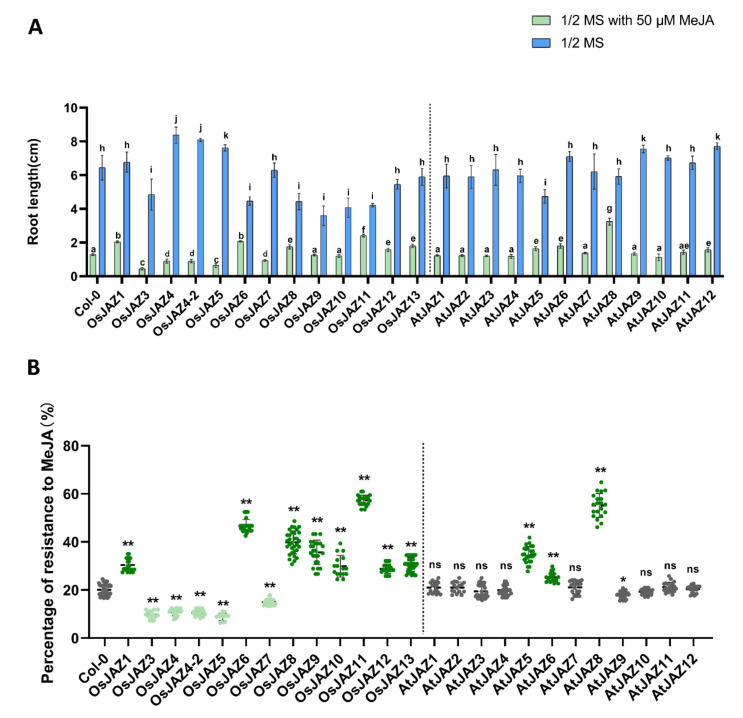
Function of JAZs in root length determination. (**A**) root length of plants grown on 1/2-strength MS medium. Root length was measured after 10 days of vertical culture on 1/2-strength MS media supplemented with 50 µM MeJA (green) or without MeJA (blue). Different letters represent significant differences among *JAZ* OE plants or JA treatments (*p* < 0.05). (**B**) The dark green points represent the MeJA resistance of the transgenic line being higher than that of Col-0, the light green points represent the MeJA resistance being lower than that of Col-0, and the gray points represent that there is no significant difference in the MeJA resistance of the transgenic line compared with Col-0 (except *AtJAZ 9* OE). The data shown are the means ± SEs (*n* > 12) (*, *p* < 0.05; **, *p* < 0.01; ns, no significance; Student’s *t*-test).

**Figure 3 ijms-23-04581-f003:**
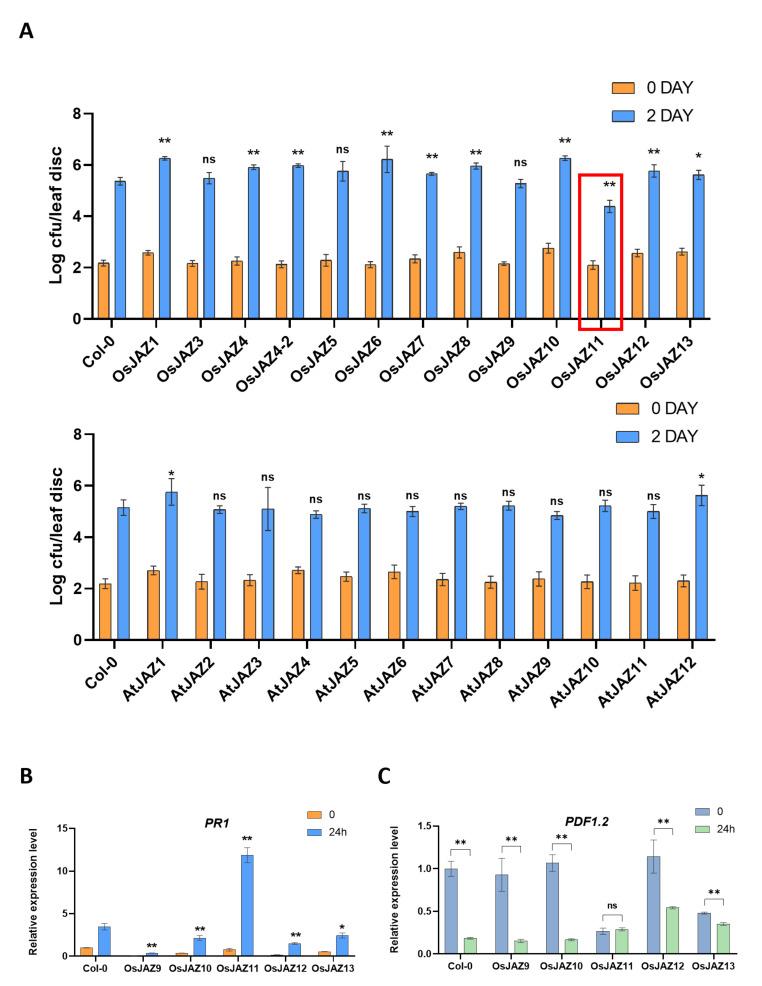
Overexpression of OsJAZ11 enhances plant resistance to *Pst* DC3000. (**A**) Three- to four-week-old plants were infiltrated with *Pst* DC3000 bacteria (1 × 10^4^ CFU·mL^−1^) via a syringe, and the bacterial population in the leaves was quantified on day 0 to confirm uniform inoculation and again two days after inoculation. The results in the graphs are the means of two independent experiments (*n* = 8 to 12) ± SEs (*, *p* < 0.05; **, *p* < 0.01; ns, no significance; Student’s *t*-test). CFU = colony forming unit. (**B**,**C**) Relative transcript levels of *PR1* and *PDF1.2* in different lines before and after inoculation. The data shown are the means ± SEs (*n* = 3) (*, *p* < 0.05; **, *p* < 0.01; ns, no significance; Student’s *t*-test).

**Figure 4 ijms-23-04581-f004:**
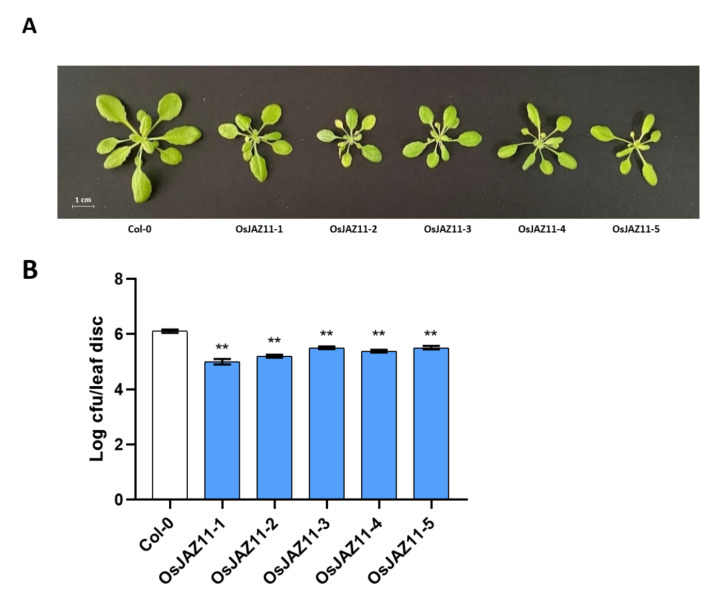
Different lines of *OsJAZ11* OE are smaller than Col-0 and have stronger resistance to *Pst* DC3000. (**A**) phenotypic profiles of Col-0 and five different lines of *OsJAZ11* OE plants grown for 25 days. The *Arabidopsis thaliana* plants were grown in a growth chamber at 22 °C under 16 h of light (100 μE·m^−2^·s^−1^) and 8 h of darkness. (**B**) Three- to four-week-old plants were syringe-infiltrated with the *Pst* DC3000 bacterium (1 × 10^4^ CFU·mL^−1^) and the bacterial population in the leaves was enumerated at two days after inoculation. Data shown are the means ± SE (*n* = 12) (** *p* < 0.01, Student’s *t*-test).

**Figure 5 ijms-23-04581-f005:**
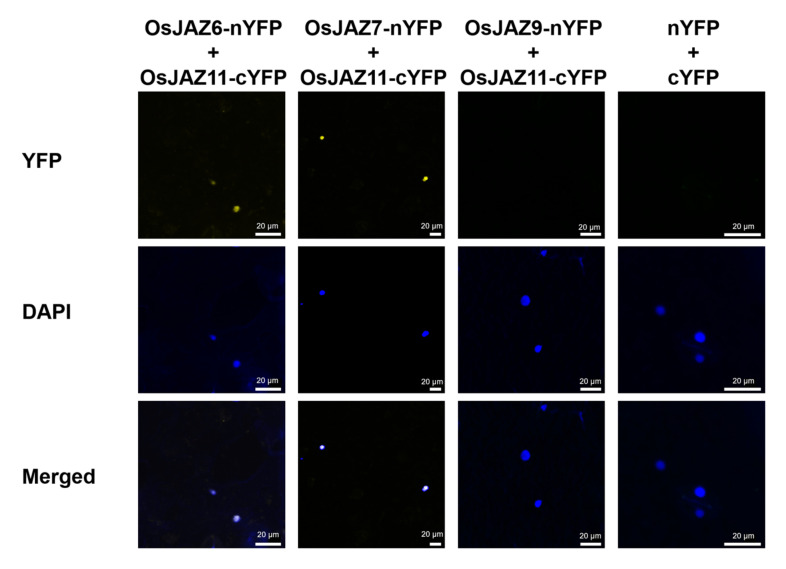
Heteromeric interaction of rice JAZ proteins. BiFC assay of heteromeric interactions in planta. YFP fluorescence was detected in *Nicotiana benthamiana* leaves coinfiltrated with *Agrobacterium* strains expressing OsJAZ6-nYFP, OsJAZ7-nYFP and OsJAZ11-cYFP. DAPI staining showed the location of the nuclei. The merged image shows the colocalization of DAPI and YFP fluorescence.

**Figure 6 ijms-23-04581-f006:**
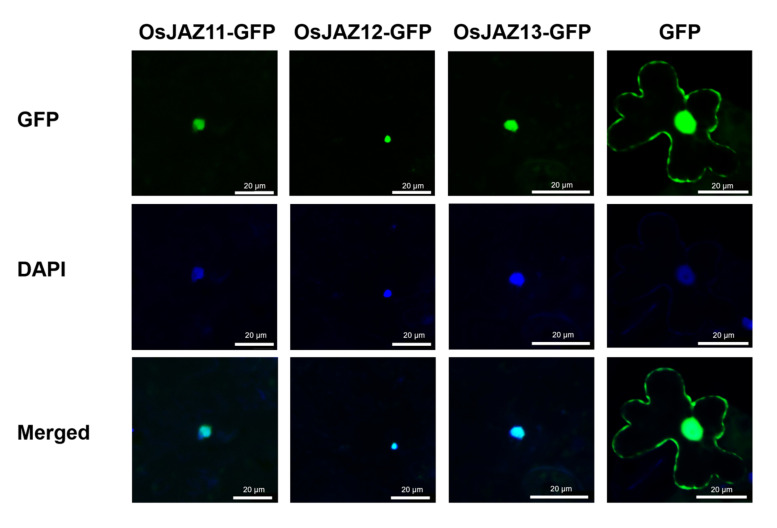
Subcellular protein localization. The fluorescence of the JAZ-GFP fusion protein in the OE plants in the nucleus was observed under a microscope. GFP-overexpressing transgenic plants were used as controls.

**Figure 7 ijms-23-04581-f007:**
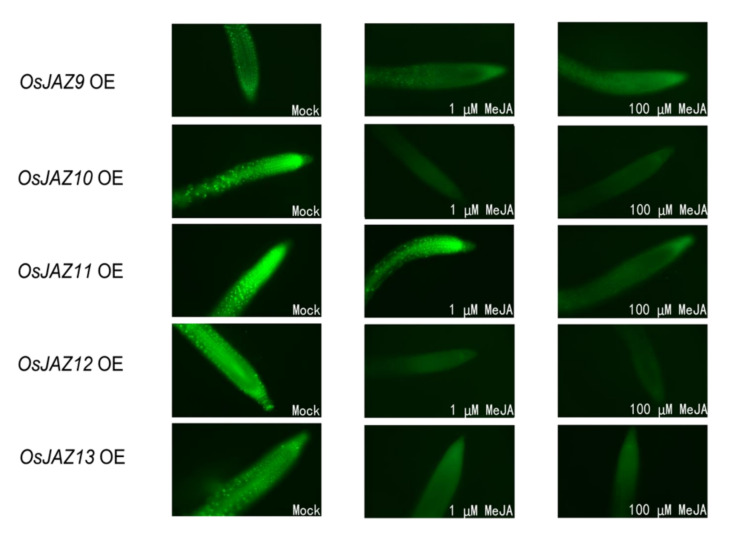
Hormone-dependent stability of OsJAZ9–12 in vivo. The root tips of transgenic seedlings were treated with water or subjected to the specified concentration of MeJA for two hours, after which the fluorescence intensity of the OsJAZ-GFP fusion protein was observed under a microscope. The exposure times for each of the above images were identical.

**Table 1 ijms-23-04581-t001:** Summary of OsJAZ–OsJAZ interactions in yeast. Each of the 13 full-length rice JAZs was tested in both AD (top row) and BD (left column) orientations. Based on the intensity of colony growth, the strength of each interaction was rated as strong (++), weak (+), or undetectable (-), as shown in Appendix A.

A D	OsJAZ1	OsJAZ3	OsJAZ4	OsJAZ4-2	OsJAZ5	OsJAZ6	OsJAZ7	OsJAZ8	OsJAZ9	OsJAZ10	OsJAZ11	OsJAZ12	OsJAZ13
B D													
**OsJAZ1**	-	-	-	-	-	-	-	-	++	++	++	-	-
**OsJAZ3**	-	-	-	-	-	-	-	-	-	++	-	-	-
**OsJAZ4**	-	-	-	-	-	-	-	-	++	++	++	++	-
**OsJAZ4-2**	-	++	++	++	++	++	++	-	++	++	++	++	-
**OsJAZ5**	-	-	-	-	-	-	-	-	-	-	-	-	-
**OsJAZ6**	+	-	-	-	++	-	++	-	++	++	++	-	-
**OsJAZ7**	-	-	-	+	++	-	++	+	++	++	++	++	-
**OsJAZ8**	-	-	-	-	-	-	-	-	-	-	-	-	-
**OsJAZ9**	-	-		-	-	-	-	-	++	-	-	-	-
**OsJAZ10**	-	-	+	-	++	-	-	-	++	++	-	++	+
**OsJAZ11**	++	-	-	++	++	++	++	-	-	++	++	-	+
**OsJAZ12**	-	-	-	++	-	-	-	++	-	-	-	-	-
**OsJAZ13**	-	++	++	++	-	-	++	++	-	+	++	-	-

**Table 2 ijms-23-04581-t002:** Summary of the *JAZ* OE phenotype and JAZ proteins homology. ”I”, ”II”, ”III” represent the group of JAZ proteins in the evolutionary tree. The dark red fills represents that the transgenic line flowered later than Col-0, the light red fills represents that the transgenic line flowered earlier than Col-0; The dark green fills represents that the MeJA resistance of the transgenic line is higher than that of Col-0, the light green fills represents that the MeJA resistance is lower than that of Col-0; The dark blue fills represents that the bacterial resistance of the transgenic line is higher than that of Col-0, the light blue fills represents that the bacterial resistance is lower than that of Col-0; All white fills represent that the phenotype of the transgenic line is not significantly different from Col-0. The black box represents that the JAZ protein in it is located on the smallest branch of the evolutionary tree.

	Phylogenetic Group	Days to Flowering	MeJA Resistance	Bacterial Resistance
**OsJAZ8**	**I**			
**OsJAZ6**	**I**			
**OsJAZ7**	**I**			
**OsJAZ11**	**I**			
**OsJAZ12**	**I**			
**OsJAZ9**	**I**			
**OsJAZ10**	**I**			
**OsJAZ13**	**I**			
**OsJAZ1**	**II**			
**OsJAZ3**	**II**			
**OsJAZ4**	**II**			
**OsJAZ5**	**II**			
**AtJAZ10**	**II**			
**AtJAZ3**	**II**			
**AtJAZ4**	**II**			
**AtJAZ9**	**II**			
**AtJAZ11**	**II**			
**AtJAZ12**	**II**			
**AtJAZ1**	**I**			
**AtJAZ2**	**I**			
**AtJAZ5**	**I**			
**AtJAZ6**	**I**			
**AtJAZ7**	**III**			
**AtJAZ8**	**III**			

## Data Availability

All data in the present research are available in the public database as referred in the Section 4.

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
