# Peer review of "Ectopic Expression of OsJAZs Alters Plant Defense and Development"

_ijms, 2022, doi:10.3390/ijms23094581_

Round 1

Reviewer 1 Report

In the manuscript authors study the overexpression of rice and Arabidopsis JAZ family proteins in Arabidopsis plants and their implication in flowering, root length and immunity against the bacteria Pst DC3000. They highlight the importance of OsJAZ11 in Pst resistance.

The manuscript is well written so it is clear and easy to read, the experiments are well designed and the results and conclusions are coherent with the aim of the study.

I have some comments and suggestions that might help to improve the paper:

-Line 104 (2.1 title). I suggest to change this title, since from the phylogenetic tree constructed in figure 1 one can not tell how closely related OsJAZs and AtJAZs are, and the statement “relatively closed” is not accurate. Study of phylogenetic relationships between OsJAZs and AtJAZs or something similar could be a more general title. The same can be applied to the statement “Some OsJAZs showed very close homologous relationships with AtJAZs”, line 27 from abstract. 

-Line 106. The use of dash (-) is ambiguous, better to use comma (,)

-Line 106- Why do you include in the study 13 out of the 15 OsJAZ members? Why are not all Arabidopsis members included?

-Your results show that OsJAZ11 is the most MeJA resistant and stable protein, which could explain resistance to coronatine degradation and protection of Arabidopsis plants expressing ectopically OsJAZ11 to Pst. What is your opinion about the fact that this resistance mechanism described for OsJAZ11 is not present in Arabidopsis in any of the AtJAZ?

-Line 109. somewhat homologous is an inaccurate expression

-Line 110. What do you mean with smallest branch of the evolutionary tree?

-Line 125. Please complete Figure 1 caption. Add if values indicated on the branches correspond to bootstrap values, if the symbols next to each taxon correspond to predicted motifs, what is X axis of the plot, what are the differences between the two motif plots (left and right) and how have you predicted those motifs.

-Line 125. In figure caption, you indicate that the tree was built by the maximum parsimony method and in line 471 of material and methods you indicate that was built by neighbor-joining method. 

-Indicate in Figure 1 which are groups I, II and III that you mention in the manuscript.

-Line 139. counted the number rosette leaves

-Line 146. I suggest to add in brackets what FT stands for, eg: FT (Flowering locus T)

-Line 225. Do you mean the importance of JA signaling in pathogen defense in rice?

-Line 264. LexA (uppercase A)

-Line 306. Extra period (.)

-Line 308. Extra “still”

-Line 308. “they are significant differences in root length (Figure 3A), bacterial resistance (Figure 4A)” wrong verb, missing connector.

-Line 335. With “Changed” you mean are different?

-Line 338. “Are” instead or “is”

-Indicate how you calculate the percentage of resistance to MeJA (figure 3B) in materials and methods.

-Line 346. Earlier

-Line 364. Wild type

-Line 386. What do you mean with “the same family”?

-Line 421. I suggest to change "evolutionary tree” in table 2  to “phylogenetic group”.

Author Response

Dear Academic Editor,

Thanks very much for taking your time to review this manuscript. These comments are all valuable and extremely helpful for revising and improving our paper, as well as the important guiding significance to our research. We have carefully studied the comments and made corrections. Please find my itemized responses in below and my revisions in the re-submitted files.

Q1: -Line 104 (2.1 title). I suggest to change this title, since from the phylogenetic tree constructed in figure 1 one can not tell how closely related OsJAZs and AtJAZs are, and the statement “relatively closed” is not accurate. Study of phylogenetic relationships between OsJAZs and AtJAZs or something similar could be a more general title. The same can be applied to the statement “Some OsJAZs showed very close homologous relationships with AtJAZs”, line 27 from abstract. 

R1: Thanks for pointing out the question. We have changed the title of this paragraph on line 104, and deleted inappropriate sentences in line 27, please check.

Q2: -Line 106. The use of dash (-) is ambiguous, better to use comma (,)

R2: Thanks for the suggestion. We have made a change in line 106, please check.

Q3: -Line 106- Why do you include in the study 13 out of the 15 OsJAZ members? Why are not all Arabidopsis members included?

R3: Thanks for your question. Some OsJAZ OE lines were missing because we did not clone the corresponding gene fragments. All 12 AtJAZ-overexpressing lines were included in this study.

Q4: -Your results show that OsJAZ11 is the most MeJA resistant and stable protein, which could explain resistance to coronatine degradation and protection of Arabidopsis plants expressing ectopically OsJAZ11 to Pst. What is your opinion about the fact that this resistance mechanism described for OsJAZ11 is not present in Arabidopsis in any of the AtJAZ?

R4: Based on the results in this manuscript, we speculate that there may be differences in the mechanism of JAZ family regulation of JA among different species. It is also possible that a special domain in the OsJAZ11 protein is responsible for this phenomenon, which is the focus of our next study.

Q5: -Line 109. somewhat homologous is an inaccurate expression

R5: Thank you for pointing out the problem. We have removed the inappropriate language on line 109.

Q6: -Line 110. What do you mean with smallest branch of the evolutionary tree?

R6: The smallest branch represents the cluster formed by the two JAZ genes with the highest sequence homology.

Q7: -Line 125. Please complete Figure 1 caption. Add if values indicated on the branches correspond to bootstrap values, if the symbols next to each taxon correspond to predicted motifs, what is X axis of the plot, what are the differences between the two motif plots (left and right) and how have you predicted those motifs.

R7: Thanks for the suggestion. We moved the phylogenetic tree to Supplementary Materials as Figure S1 and updated the legend. Added descriptions for bootstrap values ​​on branches, symbols for motifs, x-axis.

Q8: -Line 125. In figure caption, you indicate that the tree was built by the maximum parsimony method and in line 471 of material and methods you indicate that was built by neighbor-joining method. 

R8: Thanks for pointing out the problem. We have made changes in the legend of Figure S1.

Q9: -Line 139. counted the number rosette leaves

R9: Thanks for pointing out the problem. Please check line 135, we have made corrections.

Q10: -Line 146. I suggest to add in brackets what FT stands for, eg: FT (Flowering locus T)

R10: Thanks for the suggestion, we have added the description as you suggested, please check it on line 141.

Q11: -Line 225. Do you mean the importance of JA signaling in pathogen defense in rice?

R11: Thank you for pointing out the problem, we have made changes on line 220 to make the presentation clearer.

Q12: -Line 264. LexA (uppercase A)

R12: Thanks for pointing out the problem. We have made corrections on line 259.

Q13: -Line 306. Extra period (.)

R13: Thanks for pointing out the question. We have removed redundant symbols on line 301.

Q14: -Line 308. Extra “still”

R14: Thanks for pointing out the problem. We have removed redundant words on line 303.

Q15: -Line 308. “they are significant differences in root length (Figure 3A), bacterial resistance (Figure 4A)” wrong verb, missing connector.

R15: Thanks for pointing out the grammar mistake. We have made corrections on lines 303-304.

Q16: -Line 335. With “Changed” you mean are different?

R16: Yes, we have replaced with more appropriate words on line 387.

Q17: -Line 338. “Are” instead or “is”

R17: Thanks for pointing out the grammar problem. We made a correction on line 390.

Q19: -Indicate how you calculate the percentage of resistance to MeJA (figure 3B) in materials and methods.

R19: Thanks for the suggestion. We have added descriptions in “Materials and Methods” on lines 515 to 518.

Q20: -Line 346. Earlier

R20: Thanks for pointing out the problem. We have made corrections on line 399.

Q21: -Line 364. Wild type

R21: Thanks for pointing out the problem. We have made corrections on line 416.

Q22: -Line 386. What do you mean with “the same family”?

R22: Thanks for pointing out the problem. We've corrected this to “the same cluster” at line 438.

Q23: -Line 421. I suggest to change "evolutionary tree” in table 2  to “phylogenetic group”.

R23: Thanks for the suggestion. We have made changes in Table 2.

Reviewer 2 Report

Hereunder are my remarks that, in my opinion, can improve the manuscript: 

The originality of the work is missing at the end of the introduction. Please define and present clearly the aim of your study by using for instance “This study aimed to ….”.

Please change the format of supplementary files from tif to word or pdf format. 

Titles of figures in the supplementary files need to be adapted. We cannot put “supplementary figure..” in titles.

It would be better to transfer Table 2 to the results section. We cannot have tables like that in the discussion part.

Materials & methods: The section “4.12. accession numbers” need to be combined with plant material.  Some accessions numbers are in grey and other in blue (due to the attached link), why don’t you link links to all accessions? Maybe it would be better to move them to supplementary files or to “data availability” after the text.

Why didn’t you present the phylogenetic tree in the  supplementary files?

Please support your discussion with more published works.

Conclusions are lacking.

Good luck,

Author Response

Dear Academic Editor,

Thanks very much for taking your time to review this manuscript. We really appreciate all your generous comments and suggestions! Please find my itemized responses in below and my revisions in the re-submitted files.

Q1: The originality of the work is missing at the end of the introduction. Please define and present clearly the aim of your study by using for instance “This study aimed to ….”.

R1: Thanks for the suggestion. We have supplemented on lines 32-35.

Q2: Please change the format of supplementary files from tif to word or pdf format. 

Titles of figures in the supplementary files need to be adapted. We cannot put “supplementary figure..” in titles.

R2: Thanks for the suggestion. We have corrected the supplementary document as you suggested, please check it out.

Q3: It would be better to transfer Table 2 to the results section. We cannot have tables like that in the discussion part.

R3: Thanks for the suggestion. Thanks for the suggestion. We have moved Table 2 to the end of the results section, please check at lines 338-365.

Q4: Materials & methods: The section “4.12. accession numbers” need to be combined with plant material.  Some accessions numbers are in grey and other in blue (due to the attached link), why don’t you link links to all accessions? Maybe it would be better to move them to supplementary files or to “data availability” after the text.

R4: Thanks for pointing out the question. We have combined accession numbers with plant material,link links to all accessions,and move them to supplementary Table S3.

Q5: Why didn’t you present the phylogenetic tree in the supplementary files?

R5: Thanks for the suggestion. We have moved the phylogenetic tree to Supplementary as Figure S1.

Q6: Please support your discussion with more published works.

R6: Thanks for the suggestion. We have added more citations to the discussion.

Q7: Conclusions are lacking.

R7: Thanks for the suggestion. We added a conclusion at the beginning of the discussion, please check on lines 371 to 379.

This manuscript is a resubmission of an earlier submission. The following is a list of the peer review reports and author responses from that submission.

Round 1

Reviewer 1 Report

This manuscript compares the functions of JAZ transcriptional repressors in rice and Arabidopsis, through ectopic expression of OsJAZs and overexpression of AtJAZs in Arabidopsis thaliana. Overall, this study is biologically significant, because it focuses on JA signaling which is a key pathway to regulate plant growth and development. The story is well organized including both the bioinformatics analysis and experiment designs. The manuscript is clearly written. This study improves our knowledge of JAZs’ roles in rice, an important crop. It will benefit the manuscript if some concerns can be addressed.

  1. In Figure 2 and 3, it is not clear why the different color ranges are used in panels 2A, 2B and 2C. For example, in panel 2A, what is the dark red column for? It will be better to clarify it in the legend.
  2. In Figure 2D, the font size looks quite smaller than other panels. It will be better to keep the similar/same font style with other panels.
  3. In Figure 2-5, it is better to clarify what the “ns” represent in the legends, even though some experts already know it?
  4. In Figure 5A, the scale bar should be added.
  5. In Table 1, the name of JAZ family member, it looks like it starts from the number “0”/”zero” but not the letter “O”. In addition, it is better to put the gene OsJAZ4-2 in the same line instead of separating into two lines.
  6. The text style in all figures should be consistent, for example, only the first letter of the first word in a sentence/title/axis label is capitalized.

Author Response

Dear reviewer,

Thank you very much for taking your time to review my manuscript “Ectopic expression of OsJAZs alters plant defense and development”. Thank you for your recognition of our work! We really appreciate all your generous comments and suggestions! Please find my itemized responses in below and my revisions in the re-submitted files.

Reviewer

Q1: In Figure 2 and 3, it is not clear why the different color ranges are used in panels 2A, 2B and 2C. For example, in panel 2A, what is the dark red column for? It will be better to clarify it in the legend.

Response:We are grateful for the suggestion. I have indicated what each color represents in the legends of Figure 2(lines 167-178) and Figure 3(lines 201-219), please check.

Q2: In Figure 2D, the font size looks quite smaller than other panels. It will be better to keep the similar/same font style with other panels.

Response:Thank you for your suggestion. I have put Figure 2D as a Supplemental Figure S2 and used a new analysis method to make it clear and understandable.

Q3: In Figure 2-5, it is better to clarify what the “ns” represent in the legends, even though some experts already know it?

Response:Thank you for your suggestion. I have indicated what it stands for in all places where “ns” is involved, please check (line 170,175,179,206,210,242,244).

Q4: In Figure 5A, the scale bar should be added.

Response:Thank you for your suggestion. I have added a scale bar to figure 5A, please check.

Q5: In Table 1, the name of JAZ family member, it looks like it starts from the number “0”/”zero” but not the letter “O”. In addition, it is better to put the gene OsJAZ4-2 in the same line instead of separating into two lines.

Response:Thank you for your precious comments and advice. I have modified according to your suggestion to make Table 1 more beautiful.

Q6: The text style in all figures should be consistent, for example, only the first letter of the first word in a sentence/title/axis label is capitalized.

Response:We are grateful for the suggestion. We have unified the heading styles in all pictures according to your suggestions.

Reviewer 2 Report

Sun et al. sought to elucidate the function of rice JAZ transcriptional repressors and compared these to Arabidopsis JAZ proteins. The authors used both OsJAZ and AtJAZ overexpression Arabidopsis lines to look at their effect on commonly known JA-related growth and defense-related biological processes. For a subset of the proteins, the authors confirmed nuclear sublocalization and that both homo- and heterodimeric interactions existed between certain OsJAZ repressors. OsJAZ11 was investigated further and was found to be the only JAZ to confer resistance to Pst DC3000 and had elevated SA (PR1) signaling with low JA (PDF1.2) signaling, and had strongly regulated JA repression even under JA treatment. Overall, the manuscript is well-written, explores an important aspect of plant biology, and provides valuable functional characterization of an important family of transcriptional repressors.

Some minor revisions would help clean up or clarify some points in the manuscript:

  1. The JAZs were characterized in Arabidopsis overexpression lines. Why not conduct OsJAZ overexpression in rice?
  2. Figure 2A: Y axis should likely be labeled "days to flowering" not "days of flowering"
  3. Figure 2D: Is it possible to make Fig 2D larger? Text is too small to read. Also, X axis is labeled "level of flowering time" What does that mean? Days after planting? Days after earliest flowering?
  4. Line 181: insert a dash in front of OsJAZ13 (“-13”) and in front of AtJAZ12 (“-12”)
  5. Line 188: the word “media” is repeated; only one is necessary
  6. Lines 209-210: “and we found that all five OsJaz11 OE lines were slightly smaller than Col-0…” Were you concerned that less surface area=less bacteria? Did you take a uniform segment from each leaf and quantify the bacteria present in that sample? Also, in your methods section, it appears some plants were infiltrated and some were sprayed, which lines got which treatment? Could that explain differences among some of the OE lines?
  7. Figure 5: Why is fig 5A grouped with 5B and 5C? They don’t appear to be related. I would combine Fig. 5A panel with supplemental figure S3. Then combine Fig 4A/B panels with 5B/5C panels (doing this would generate a new figure 4 with four panels to show resistance levels in (A) OsJAZs, (B) AtJAZs, and (C:PR1/D:PDF1.2) explain the underlying molecular mechanisms via qRT-PCR for a subset of the OsJAZ OE lines. Just a suggestion but it seems like it would flow better and it would require very minimal changes to the text.
  8. Starting with PR1/PDF1.2 expression results, only a subset of JAZs (OsJAZ9-OsJAZ13) are included in some of the experiments. Can you include a short explanation of why you chose to do further experiments with those specific JAZ OE lines?
  9. Line 248: Would be helpful to tell reader JAZ9-JAZ11 interaction was included as a negative test to justify why you were using that one. Also, Fig S4: Can you rotate the horizontal text vertical and rotate the vertical text horizontal? I think the text would line up a little better with the rows and columns.
  10. Line 358: “obviously essentially” wording sounds odd. Do you need both words?
  11. Lines 363-364: "which may be due to the reduction in functional redundancy between JAZs caused by ectopic expression.” I'm not sure I understand this statement. Are you saying that ectopic expression of OsJAZs may cause an increase in functionalities such that immune functions are altered in those plants? Could you clarify what you mean?
  12. Lines 380-381: Either before or after your statement that you speculate the resistance in OsJAZ11 is due to JA/SA antagonism, it would be beneficial for your argument to mention a biological example that parallels your results: Previous research has shown that the SA pathway is involved in resistance against Pst DC3000. Furthermore, Pst DC3000 produces coronatine, a structural mimic of bioactive JA that is more potent than jasmonates in activating JA responses (Fonesca et al., 2009) and coronatine also downregulates SA respones (Brooks et al., 2005) Zheng et al., 2012). By overexpressing the strong and stable repressor OsJAZ11, these plants are presumably able to withstand degradation of the JAZs after coronatine exposure. Therefore, JA response is not increased (i.e. as shown by low PDF1.2 levels) and the plant is able to induce the SA response (marked by increased PR1)  and mounts a defense response that results in increased resistance to Pst DC3000.

Fonseca, S., Chini, A., Hamberg, M. et al. (+)-7-iso-Jasmonoyl-L-isoleucine is the endogenous bioactive jasmonate. Nat Chem Biol 5, 344–350 (2009). https://doi.org/10.1038/nchembio.161

Brooks et al., 2005: The Pseudomonas syringae phytotoxin coronatine promotes virulence by overcoming salicylic acid-dependent defences in Arabidopsis thaliana. Mol Plant Pathol. 2005 Nov 1;6(6):629-39. doi: 10.1111/j.1364-3703.2005.00311.x.

Zheng et al., 2012: Coronatine Promotes Pseudomonas syringae Virulence in Plants by Activating a Signaling Cascade that Inhibits Salicylic Acid Accumulation. Cell Host and Microbe. Volume 11, ISSUE 6, P587-596, June 14, 2012 DOI: https://doi.org/10.1016/j.chom.2012.04.014

Author Response

Dear reviewer,

Thanks very much for taking your time to review this manuscript. We really appreciate all your generous comments and suggestions! Your serious and rigorous attitude left a deep impression on me, and it is worth learning from me. Please find my itemized responses in below and my revisions in the re-submitted files.

Reviewer

Q1: The JAZs were characterized in Arabidopsis overexpression lines. Why not conduct OsJAZ overexpression in rice?

Response:Thank you for reminding me. In this study, we wanted to compare the similarities and differences between JAZ proteins in rice and arabidopsis, and the arabidopsis system is more efficient and easier to handle. Based on the study of Arabidopsis, we will select more valuable genes for further study in rice.

Q2: Figure 2A: Y axis should likely be labeled "days to flowering" not "days of flowering"

Response:Thanks for pointing out my grammatical error. I have modified it according to your suggestion, please check.

Q3: 3. Figure 2D: Is it possible to make Fig 2D larger? Text is too small to read. Also, X axis is labeled "level of flowering time" What does that mean? Days after planting? Days after earliest flowering?

Response:Thank you for your suggestion. I have put Figure 2D as Supplemental Figure S2 and used a new analysis method to make it clear and understandable, please check.

Q4: Line 181: insert a dash in front of OsJAZ13 (“-13”) and in front of AtJAZ12 (“-12”)

Response:Thank you for your suggestion. I have modified it according to your suggestion, please check (lines 187-188).

Q5: Line 188: the word “media” is repeated; only one is necessary

Response:Thank you for pointing out my mistake, I have deleted the repeated words.

Q6:  Lines 209-210: “and we found that all five OsJaz11 OE lines were slightly smaller than Col-0…” Were you concerned that less surface area=less bacteria? Did you take a uniform segment from each leaf and quantify the bacteria present in that sample? Also, in your methods section, it appears some plants were infiltrated and some were sprayed, which lines got which treatment? Could that explain differences among some of the OE lines?

Response:When testing the resistance of transgenic plants to Pst DC3000, we use a bacterial suspension with OD=0.002 to completely inject the whole leaf. After leaf spots appear (2-3 days), cut out leaves of approximately the same weight from each line, and measure their weight with a microbalance to quantify the bacterial count. So, our experimental results are credible. During the experiment, we used two methods, spraying and injection, to ensure that the results are reproducible. All relevant pictures in the article are the result of injection.To avoid misunderstandings, we deleted the descriptions related to spraying.

Q7: Figure 5: Why is fig 5A grouped with 5B and 5C? They don’t appear to be related. I would combine Fig. 5A panel with supplemental figure S3. Then combine Fig 4A/B panels with 5B/5C panels (doing this would generate a new figure 4 with four panels to show resistance levels in (A) OsJAZs, (B) AtJAZs, and (C:PR1/D:PDF1.2) explain the underlying molecular mechanisms via qRT-PCR for a subset of the OsJAZ OE lines. Just a suggestion but it seems like it would flow better and it would require very minimal changes to the text.

Response:Thank you for your precious comments and advice. I have modified it according to your suggestion, please check.

Q8: Starting with PR1/PDF1.2 expression results, only a subset of JAZs (OsJAZ9-OsJAZ13) are included in some of the experiments. Can you include a short explanation of why you chose to do further experiments with those specific JAZ OE lines?

Response:Thank you for your suggestion. It is hard to analyze all transgenic lines and proteins, so we have selected OsJAZ9 OE-OsJAZ11 OE for analysis in many experiments. First, this JAZ subset contains the transgenic plant OsJAZ11 OE that we care about most. Secondly, this subset is representative in many phenotypes. In many indicators, there was an increase, a decrease or no significant difference relative to Col-0. And based on your suggestion, we added a description in the original text, please check (lines 224-225).

Q9: 9. Line 248: Would be helpful to tell reader JAZ9-JAZ11 interaction was included as a negative test to justify why you were using that one. Also, Fig S4: Can you rotate the horizontal text vertical and rotate the vertical text horizontal? I think the text would line up a little better with the rows and columns.

Response:We are grateful for the suggestion. I have modified it according to your suggestion, please check.

Q10: Line 358: “obviously essentially” wording sounds odd. Do you need both words?

Response:Thank you for reminding me. I deleted these two words to make the sentence more fluent, please check(lines 383-386).

Q11: Lines 363-364: "which may be due to the reduction in functional redundancy between JAZs caused by ectopic expression.” I'm not sure I understand this statement. Are you saying that ectopic expression of OsJAZs may cause an increase in functionalities such that immune functions are altered in those plants? Could you clarify what you mean?

Response:Thank you for your question. Our statement is incorrect, and we have revised it(line 390). Only when gene knockout and suppression of expression exist, there is a problem of functional redundancy, and there is no problem of functional redundancy in overexpression.

Q12: Lines 380-381: Either before or after your statement that you speculate the resistance in OsJAZ11 is due to JA/SA antagonism, it would be beneficial for your argument to mention a biological example that parallels your results: Previous research has shown that the SA pathway is involved in resistance against Pst DC3000. Furthermore, Pst DC3000 produces coronatine, a structural mimic of bioactive JA that is more potent than jasmonates in activating JA responses (Fonesca et al., 2009) and coronatine also downregulates SA response (Brooks et al., 2005) Zheng et al., 2012). By overexpressing the strong and stable repressor OsJAZ11, these plants are presumably able to withstand degradation of the JAZs after coronatine exposure. Therefore, JA response is not increased (i.e. as shown by low PDF1.2 levels) and the plant is able to induce the SA response (marked by increased PR1)  and mounts a defense response that results in increased resistance to Pst DC3000.

Response:Thanks for your suggestions, it makes this article more logical. I have slightly modified the example sentence you gave and added it to the article, please check (lines 220-230, lines 404-412).